# OpenReview forum: "Learning Diverse Quadruped Locomotion Gaits via Reward Machines"
_ICLR.cc/2024/Conference — ICLR 2024 Conference Withdrawn Submission_

### Official Review · Reviewer_i6Hh · 2023-10-26

**Soundness:** 4 excellent
**Presentation:** 4 excellent
**Contribution:** 3 good
**Rating:** 8
**Confidence:** 4

**Summary:**

This paper proposes a reward machine to allow learning different quadrupedal gaits for quadrupedal robots. Multiple gaits are learned for a real quadrupedal robot, including novel gaits such as Three-One.

**Strengths:**

1. A novel way to encourage policies to produce different gaits for quadrupedal robots. Including the learning of novel gaits.

2. Good ablation studies to evaluate the importance of different components.

**Weaknesses:**

It will be nice to also learn transitions between gaits.

**Questions:**

It will be interesting to evaluate energy consumption for different gaits at different speeds, e.g., one will expect certain gaits to be more energy efficient at high speed while less at low speed. It will also be fun to try out gaits that are typical at high speed in nature, like galloping, even if only demonstrated in simulation.

---

> ### Author Response · Authors · 2023-11-15
>
> We respond to each comment below.
>
> > 1. A novel way to encourage policies to produce different gaits for quadrupedal robots. Including the learning of novel gaits.
> > 2. Good ablation studies to evaluate the importance of different components.
>
> We thank the reviewer for the positive comments.
>
> > It will be nice to also learn transitions between gaits.
>
> Transitioning between locomotion gaits is an interesting open problem in legged locomotion. We believe that transitioning between different reward machines can be a useful contribution to this problem. We mention this as a future direction in section 5: “...nor have we studied how to smoothly transition between gaits”. We feel this is better suited for future work due to the challenging nature of this problem - it requires smoothly transitioning between policies, or training a unified policy for multiple gaits.
>
> > It will be interesting to evaluate energy consumption for different gaits at different speeds, e.g., one will expect certain gaits to be more energy efficient at high speed while less at low speed. It will also be fun to try out gaits that are typical at high speed in nature, like galloping, even if only demonstrated in simulation.
>
> We agree with both of these comments. There are many dimensions to evaluate different gaits on, including speed (linear and angular), gait frequency, and terrain. Across each dimension, we can measure different variables such as energy consumption, stability, and velocity tracking accuracy. This evaluation can occur in simulation and on hardware. We can also further expand the diversity of learned gaits to include galloping, canter, and skipping to name a few.

---

### Official Review · Reviewer_6Qms · 2023-10-31

**Soundness:** 2 fair
**Presentation:** 3 good
**Contribution:** 2 fair
**Rating:** 5
**Confidence:** 4

**Summary:**

The paper introduces reward machines for learning different quadrupedal gaits called Reward Machine-based Locomotion Learning (RMLL). The key to the proposed approach is introducing high-level gait specifications as automaton states and a counter to control gait frequency. The authors construct a automaton via LTL formulas representing foot-contacts. PPO is used to learn a policy which takes the state as a combination of automaton state, frequency counter, proprioception and commands to output target joint angles. More rewards are given for transitioning to subsequent automaton states. All the learned gaits are demonstrated in the real-world.

**Strengths:**

The presented approach shows a straightforward way of learning different gaits via adding automaton structure to reward function. High-level state conditions and transitions are characterized by foot contacts and used to motivate the desired automaton state transitions. The added benefit of controlling the gait frequency at execution adds to the contributions of RMLL.

**Weaknesses:**

The proposed method’s novelty is highly limited and is more a robotics application of the base method proposed for using reward machines with RL (Icarte et al. 2018, 2019, 2022). Using automaton states with the state of the MDP has been introduced in Icarte et al. 2018. The authors introduce the timestep counter for controlling automaton state transitions.

The authors have not mentioned the exact representation of the automaton state $u$ in the input to the policy. Is it a vector of boolean? How is it different when the RM state is replaced with that of foot contacts? In that ablation, do you still keep $\phi$ for the No-RM-Foot-Contacts case?

Can you please clarify?: With foot-contacts, two consecutive states can have two different foot-contacts with random policy. However, with reward machines the RM state is constant until a transition happens? Also, in all the ablation of state space, the reward structure is based on the RM right?

Learning policy for individual gaits is limited contribution in itself. How is the energy consumption study relevant to show the efficacy of the proposed approach? If you already know which gait consumes least energy while maintaining stability as a function of the terrain, why cannot a terrain based reward machine states be formulated?


[1] Rodrigo Toro Icarte, Toryn Klassen, Richard Valenzano, and Sheila McIlraith. Using reward ma- chines for high-level task specification and decomposition in reinforcement learning. In Interna- tional Conference on Machine Learning, pp. 2107–2116. PMLR, 2018.

[2] Rodrigo Toro Icarte, Ethan Waldie, Toryn Klassen, Rick Valenzano, Margarita Castro, and Sheila McIlraith. Learning reward machines for partially observable reinforcement learning. Advances in Neural Information Processing Systems, 32:15523–15534, 2019.

[3] Rodrigo Toro Icarte, Toryn Q Klassen, Richard Valenzano, and Sheila A McIlraith. Reward ma- chines: Exploiting reward function structure in reinforcement learning. Journal of Artificial In- telligence Research, 73:173–208, 2022.

**Questions:**

See weakness above.

---

> ### Author Response · Authors · 2023-11-15
>
> We respond to each comment below.
>
> > The presented approach shows a straightforward way of learning different gaits via adding automaton structure to reward function. High-level state conditions and transitions are characterized by foot contacts and used to motivate the desired automaton state transitions. The added benefit of controlling the gait frequency at execution adds to the contributions of RMLL.
>
> We thank the reviewer for the positive comments.
>
> > The proposed method’s novelty is highly limited and is more a robotics application of the base method proposed for using reward machines with RL (Icarte et al. 2018, 2019, 2022). Using automaton states with the state of the MDP has been introduced in Icarte et al. 2018. The authors introduce the timestep counter for controlling automaton state transitions.
>
> While true that our base method relies on the existing reward machine framework, we are the first to leverage this framework for the important application of quadruped locomotion, which improves over existing quadruped locomotion approaches by specifying and learning **arbitrary** gaits through foot contact sequences **with minimal manual effort** (i.e. no motion priors or dynamics models needed). See Section 2.3 for discussion on our novelty compared with other locomotion approaches which learn diverse gaits. We leverage ideas from the machine learning community (specifically, the RM framework) to enable both easy specification and efficient learning of new gaits, which is the main contribution of this work.
>
> > The authors have not mentioned the exact representation of the automaton state $u$ in the input to the policy. Is it a vector of boolean? How is it different when the RM state is replaced with that of foot contacts? In that ablation, do you still keep $\phi$ for the No-RM-Foot-Contacts case?
>
> In Section 3.2 we mention: “The RM state is encoded as a one-hot vector, …”. When the RM state is replaced with foot contacts for “No-RM-Foot-Contacts” and “No-RM-History” ablations, we remove the RM state from the state space, and replace it with a boolean vector of foot contacts (four dimensions, one boolean value per foot). We have updated the paper (section 4.2 in blue) to make it more clear. We keep $\phi$ in the state space for all ablations, including No-RM-Foot-Contacts.
>
> > Can you please clarify?: With foot-contacts, two consecutive states can have two different foot-contacts with random policy. However, with reward machines the RM state is constant until a transition happens?
>
> We are not positive we understand what the reviewer is referring to with respect to “two consecutive states” (automata states?) and “random policy”? It is the case that the RM state is constant until a desired **gait transition** occurs. Note that the RM takes a transition at every timestep (following the RM framework in Icarte et al., 2018), however in our work we define the transition function to take a self-loop unless a desired gait transition occurs (see Figure 3, and Appendix A). In other words, we take an RM transition at every timestep, however the RM does not transition to a **new** automaton state unless the next desired gait pose is reached.
>
> > Also, in all the ablation of state space, the reward structure is based on the RM right?
>
> Each of our ablations are trained on and evaluated over the same reward function, which is based on the RM.
>
> > Learning policy for individual gaits is limited contribution in itself.
>
> We respectfully disagree. Compared to existing works, we can specify arbitrary gaits given a desired foot contact sequence (beyond two-beat gaits) with minimal human efforts (no motion priors or dynamics models). To the best of our knowledge, no existing methods support this capability.
>
> > How is the energy consumption study relevant to show the efficacy of the proposed approach? If you already know which gait consumes least energy while maintaining stability as a function of the terrain, why cannot a terrain based reward machine states be formulated?
>
> The purpose of the energy consumption study is to motivate the reason for learning diverse locomotion gaits at all. According to the results of the study, different gaits are beneficial in terms of stability and energy consumption based on terrain type. Thus, learning diverse quadruped locomotion gaits is an important problem. We do not already know “which gait consumes least energy while maintaining stability as a function of the terrain” before the study is conducted. While possible to create terrain-based reward machines, it is unclear to us what their objective is. Also, we would need to somehow estimate the terrain conditions on the real robot. It is an interesting direction, but out of the scope of our current work.

---

### Official Review · Reviewer_UMdH · 2023-11-05

**Soundness:** 2 fair
**Presentation:** 2 fair
**Contribution:** 1 poor
**Rating:** 3
**Confidence:** 5

**Summary:**

In this paper, the authors introduce reward machine, a state machine based mechanism to shape complex, state/time dependent rewards for dynamic locomotion control problems. For each desired quadruped gait, an automata is constructed to modulate foot contact transitions and timings. Then, the automata state, proprioceptive state (including estimations) from the robot, as well as gait parameters are used as the state vector for reinforcement learning training. The authors train a few different gaits in simulation and transfer the policies to the real hardware.

**Strengths:**

The strengths of the paper include:

1) Introduction of the reward machine for locomotion control and gait specification.
2) Sim2real transfer of learned policies to the real robot

**Weaknesses:**

The weaknesses of the paper are:

1) While the concept of the reward machine is new especially in the locomotion learning community, in reality it is merely a fancy way of constructing a state machine which controls the gait transition.
2) The tasks in this paper are not novel. I see only flat terrain locomotion with a few gaits, and it is hard to justify why a complex state machine is needed, given there are works that can also achieve diverse gaits with time based rewards: "Walk These Ways: Tuning Robot Control for Generalization with Multiplicity of Behavior".
3) Other than that, the learning is conducted in Isaac Gym with PPO and there is limited novelty.

**Questions:**

N/A

---

> ### Author Response · Authors · 2023-11-15
>
> We respond to each comment below.
>
> > In this paper, the authors introduce reward machine, a state machine based mechanism to shape complex, state/time dependent rewards for dynamic locomotion control problems.
>
> This is a major misunderstanding. This paper does not introduce reward machines. Reward machines were introduced by Icarte et al., 2018, which was cited in Section 2.1. In this paper, we introduce reward machines into the literature for easily specifying and learning diverse locomotion gaits.
>
> >The strengths of the paper include:
>
> > 1. Introduction of the reward machine for locomotion control and gait specification.
> > 2. Sim2real transfer of learned policies to the real robot
>
> We thank the reviewer for the positive comments.
>
> > The weaknesses of the paper are:
>
>   > 1.  While the concept of the reward machine is new especially in the locomotion learning community, in reality it is merely a fancy way of constructing a state machine which controls the gait transition.
>
> We are not sure what weakness the reviewer is trying to point out here. Indeed we are “constructing a state machine which controls the gait transition”. This state machine specifies arbitrary locomotion gaits over foot contact sequences, along with gait frequencies. This enables diverse gait specification and learning with minimal manual efforts compared to related works.
>
> > 2. The tasks in this paper are not novel.
>
> We specify and learn **arbitrary** gaits through foot contact sequences **with minimal manual effort** (i.e. no motion priors or dynamics models needed). While the task of learning diverse quadruped locomotion gaits certainly is not novel (there are many papers on this task throughout the years, as discussed and cited in Section 2.3), we approach the problem from a unique angle.
>
> > I see only flat terrain locomotion with a few gaits, and it is hard to justify why a complex state machine is needed, given there are works that can also achieve diverse gaits with time based rewards: "Walk These Ways: Tuning Robot Control for Generalization with Multiplicity of Behavior".
>
> We already mention how our work differs from “Walk These Ways: Tuning Robot Control for Generalization with Multiplicity of Behavior" in Section 2.3:
>
> “Other similar works involve specifying and learning diverse locomotion gaits through explicitly defining swing and stance phases per leg (Siekmann et al., 2020; Margolis & Agrawal, 2023). The former approach is designed for a bipedal robot, while the later only supports two-beat quadruped gaits. In contrast, RMLL only needs foot contact sequences (instead of leg-specific timings), and can specify and learn arbitrary quadrupedal gaits well beyond the set of two-beat gaits.”
>
> Note that in Section 2.3, we discuss how our approach differs from various other related works which also achieve diverse gaits.
>
> > 3. Other than that, the learning is conducted in Isaac Gym with PPO and there is limited novelty.
>
> We respectfully disagree that using Isaac Gym negatively affects the novelty of our approach (or other approaches in the literature). It is very common to train quadruped locomotion policies in Isaac Gym with PPO (See Agarwal et al.,2023, Zhuang et al., 2023, or “Deep whole-body control: learning a unified policy for manipulation and locomotion” for example). We do not claim to improve the Isaac Gym physics simulation environment, nor do we claim to improve PPO. We are happy to leverage an existing physics simulation environment and RL algorithm for our work, which contributes to quadruped locomotion in a different dimension.

---

### Official Review · Reviewer_MmwQ · 2023-11-11

**Soundness:** 3 good
**Presentation:** 3 good
**Contribution:** 2 fair
**Rating:** 5
**Confidence:** 3

**Summary:**

This paper proposes a reward design system via a set of state machines that consists of individual conditions on each of the foot contacts. This allows the user to specify different types of gaits easily. Diverse gait policies are trained via sim2real (in isaac gym) and deployed on hardware. The videos show gaits like jumping, trotting, running etc.

**Strengths:**

- The problem of automatic reward design is important for training more general policies
- The robot videos look good and the gaits are very diverse
- The motivation and approach are well-presented
- There is a lot of good analysis on the experiments and ablations, especially about the differences in the gaits

**Weaknesses:**

- I am unclear about the exact novelty of this approach - as it is already common in RL-based locomotion setups to use foot poses to generate different gaits.

- It would be more interesting to see how these reward machines can be used to do more complex, long-horizon tasks, like walking with multiple gaits or imitating a long reference trajectory. Specifically, how can one transition between different machines?

**Questions:**

See weaknesses

---

> ### Author Response · Authors · 2023-11-15
>
> We respond to each comment below.
>
> >Strengths:
>
> > - The problem of automatic reward design is important for training more general policies
> > - The robot videos look good and the gaits are very diverse
> > - The motivation and approach are well-presented
> > - There is a lot of good analysis on the experiments and ablations, especially about the differences in the gaits
>
> We thank the reviewer for the positive comments.
>
> > Weaknesses:
>
> > - I am unclear about the exact novelty of this approach - as it is already common in RL-based locomotion setups to use foot poses to generate different gaits.
>
> There are indeed existing RL-based locomotion setups which use foot poses to generate different gaits (we cite some of these approaches in Section 2.3). Our approach (RMLL) is a novel paradigm for gait specification, which requires less manual effort (only an automaton over the desired foot contact sequence), and can specify arbitrary gaits beyond simple two-beat gaits when compared to existing works such as Margolis & Agrawal, 2023; Tang et al., 2023. We believe our approach is novel w.r.t. the literature, and hope the reviewer appreciates our contribution.
>
> > - It would be more interesting to see how these reward machines can be used to do more complex, long-horizon tasks, like walking with multiple gaits or imitating a long reference trajectory.
>
> Although RMs are commonly used for high-level, long-horizon tasks (i.e. Toro Icarte et al., 2019), the focus of our work is on low-level locomotion. While it would be interesting to leverage RMs for long-horizon quadruped tasks, this is somewhat orthogonal to locomotion policy learning. An interesting follow-up work can involve optimally leveraging these different gaits to efficiently traverse various terrains. This follow-up approach can leverage reward machines for long-horizon tasks.
>
> > Specifically, how can one transition between different machines?
>
> Transitioning between locomotion gaits is an interesting open problem in legged locomotion. We believe that transitioning between different reward machines can be a useful contribution to this problem. We mention this as a future direction in section 5: “...nor have we studied how to smoothly transition between gaits”. We feel this is better suited for future work due to the challenging nature of this problem - it requires smoothly transitioning between policies, or training a unified policy for multiple gaits.